# Additional Criteria for Playground Impact Attenuating Sand

David Eager *, Chris Chapman, Yujie Qi, Karlos Ishac and Md Imam Hossain

Faculty of Engineering and Information, University of Technology Sydney, Broadway, NSW 2007, Australia; c.chapman123@icloud.com (C.C.); Yujie.Qi@uts.edu.au (Y.Q.); Karlos.ishac@uts.edu.au (K.I.); MdImam.Hossain@uts.edu.au (M.I.H.)
* Correspondence: David.Eager@uts.edu.au

**Abstract:** Falls within children's playgrounds result in long bone and serious injuries. To lower the likelihood and severity of injury, impact attenuating surfaces (IAS) are installed within the impact area (fall zone). There are three primary IAS materials used, namely: granulated rubber products, wood fibre products, and sand. There is a deficiency with existing IAS test methods in that they do not take account of sand degradation over time. When children use the playground, sand degradation can occur when sand produces fines and smaller particles with low sphericity and angular which fill the voids between the sand particles. These fines and smaller particles tend to bind the sand and lower its impact attenuating performance. This paper proposes an additional IAS test to eliminate sands that degrade above an established threshold rate after installation due to normal usage. IAS degradation properties of fifteen IAS sands were tested including sand particle shape, sand particle distribution, percentage fines and sand particle degradation. This accelerated ageing test method is applicable only to sands and not rubber or wood fibre IAS products. The best IAS sands were sourced from quarries located on rivers that had eroded volcanic outcrops. These sands were shown to degrade the least and had little to no fines, and their particle shape was rounded to well-rounded. The most reliable source for good quality IAS sands on these rivers was on specific bends. The sand mined at these locations consistently had a tight particle size distribution.

**Keywords:** children's playground; injury prevention; child falls; long bone injuries; serious injuries; impact attenuating surface; IAS; HIC; $g_{max}$

## 1. Introduction

Traditionally sand has been employed by engineers as select fill and used effectively as a compacted sub-base beneath pavements such as roads. This role for sand is opposite to what is required in children's playgrounds. For children's playgrounds, sand that is employed as an impact attenuating surface (IAS) must not compact and should remain uncompacted for its usable life. Moreover, IAS sand should remain uncompacted during and after rain or after the irrigation sprinklers have inadvertently doused it. It should not require raking to break a hard crust or compaction in high usage areas of the playground. Ideally, it should be self-levelling and maintenance-free except for periodic topping up and removal of litter and organic matter.

There has been very little research into the parameters that are required for IAS sand used to reduce fall-related injuries within children's playgrounds.

In 2004 Eager and Chapman published a discussion paper on playground IAS [1]. More recently Hayati et al. published a paper on the dynamic behaviour of high performance sand surfaces used in the sports industry [2]. IAS sand is used within children's playgrounds to reduce the likelihood and consequences of death and serious injuries resulting from falls at height from playground equipment. There are standards around the world for the measurement of the performance of the IAS [3–7]. These standards are effective in the measurement of the IAS performance but do not provide guidance on the selection of what makes a good IAS sand. It is important that the IAS sand is fit-for-purpose and

that it meets the minimum safety requirements and is maintained to these standards for the life of the playground equipment [8]. Davidson et al. [9] showed that the amount of energy dissipating away from or returned to a child impacting onto a playground surface will influence the risk of sustaining a fracture.

Up to 45% of accidents to children in playgrounds lead to serious injuries including concussions and bone fractures [10]. Furthermore, it is estimated that 67% of recorded playground injuries are due to falls from the equipment to the ground [11]. Gunatilaka et al. [12] suggest that two key risk factors for playground related injury are the height of equipment from which children fall and the type of surfacing they land. Therefore, it seems appropriate that action needs to be directed at reducing the potential for children to fall from playground equipment and reducing the severity of these falls given the likelihood that they will occur.

In preparation for this research, we undertook a literature review to consider different types of impact attenuation methods used in industry. The most widely applied test method for evaluating the impact attenuating performance of a playground surfacing material is to employ the methodology contained in the Australian Standard AS 4422:2016 [6], and its equivalent overseas standards [3–5] particularly the recently published ISO Technical Specification [7]. This process involves dropping an instrumented metal headform onto a sample of the material, with performance measured in terms of g-max and head injury criterion (HIC) parameters. Mack et al. [13] highlight the inadequacies of this method for loose fill materials (including sand) on the basis of the inconsistency in the results that are achieved. Following the investigation of a variety of loose fill materials, they conclude that reproducibility of the process was a key problem. Despite these inadequacies, the method remains at the forefront of material selection processes for IAS materials used in playground safety.

An objective of this research is to propose an accelerated aging test method to allow a quantitative assessment of how the sand will deteriorate over time, subject to use and external factors, in the context of children's playgrounds. A key industry to consider is the construction industry as it has a strong tradition of materials testing and supporting laboratories to conduct such testing. Specific to concreting practices, a vast array of quality control processes and quality assurance tests have been developed and implemented on a global scale. The Los Angeles Abrasion test was designed to assess the ability of concreting aggregates to resist the degradation and abrasion actions that occur during manufacturing, placing and compaction activities [14]. For many years the method has been the standard for evaluating the degradation and toughness of small size coarse aggregates with particle sizes typically greater than 1.70 mm [15]. Neville [16] notes that the Los Angeles test combines abrasion/attrition, impact and grinding actions—which are a consequence of the tumbling and dropping of the aggregate with steel spheres. Whilst this method is typically employed for the assessment of small size coarse aggregates, it is expected that application may be appropriate to finer aggregate materials including sand used to attenuate falls within children's playgrounds.

The Micro-Deval test is comparable to that of the Los Angeles Abrasion Test, however, [17] it was developed to provide a better indication of an aggregate's service life when exposed to weather and moisture. Micro-Deval methods exist for the evaluation of small size coarse aggregate and also fine aggregate including sand. Consistent with the Los Angeles Abrasion test, the aggregate sample is obtained from that portion of material retained from mechanical sieving—using sieve size 1.18 mm for small size coarse aggregate and 0.075 mm for fine aggregate [17]. In contrast to the Los Angeles test, ref. [17] the aggregate sample and steel balls are not raised and dropped by a rigid steel shelf. Rodgers [18] notes that the development of this method was driven by a need for a test that was simple, inexpensive and precise—an important consideration in any industry, particularly the playground industry which as a general rule is run on a wing and a prayer and does not have the luxury of large capital works budgets which are typical within the construction industry.

The Standard Compaction Effort method has application to both coarse and fine aggregates [19]. Figure 1 depicts the rammer and the cylindrical steel mould used in this study. An aggregate or soil sample is positioned within a cylindrical steel mould and repeatedly compacted by a rammer. Intended primarily to compact a material sample, Carter and Bentley [20] identify a key failure of the intended method when applied to sands and gravels. The tested material tends to be displaced by the impact of the rammer, as opposed to being compacted. The net effect is that degradation/deterioration is induced on the sand or gravel particles. The Authors of this paper saw this as an advantage: it would allow them to accelerate the degradation/deterioration of playground sand and quantify this property for individual sand products.

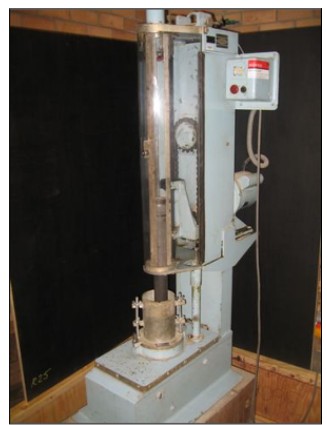 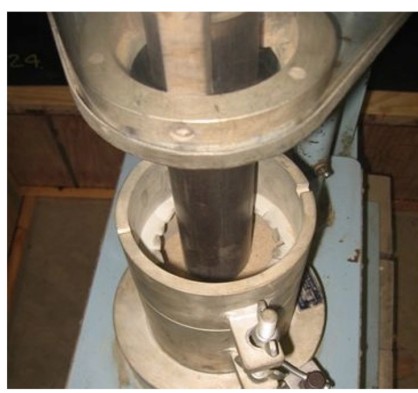

**Figure 1.** (**Left**) Mechanically operated rammer. (**Right**) Close-up of the cylindrical steel mould.

Neville [16] identifies that a common form of deterioration of aggregates is through the mechanism of freezing and thawing, which occurs by the nature of the environmental conditions in an area. The intention of freezing and thawing processes is to assess the resistance of an aggregate (or other component) to weathering and to simulate environmental effects. A freeze/thaw cycle may be defined to comprise a number of processes [21]. The material to be tested is soaked in distilled water measuring 20 °C for a period of 24 h. It is then reduced to −20 °C and maintained there for 2 h. It is then returned to 20 °C. This process is repeated a predetermined number of times. Lamond and Pielert [15] note that freezing and thawing induces potentially damaging expansion in aggregates. They further highlight that a prominent criticism of this method is its variability—both within and between different test laboratories.

Another method is the aggregate crushing value and is defined under the Australian Standard AS 1141.21—1997 Methods for Sampling and Testing Aggregates: Method 21—Aggregate Crushing Value [22]. It is a measure of the resistance an aggregate provides to crushing under the influence of a compressive load. Whilst typically applied to coarser sized aggregate, this method is applicable to finer sized aggregate including sand [22]. The sample aggregate is placed into a cylindrical steel mould where a steel plunger is used to subject the particles to a sustained compressive force over a period of time. This force should equal approximately 400 kN, and be sustained over a period of 10 min [22].

The existing IAS standards require the entire impact area (fall zone) of children's playgrounds to comply with a maximum acceleration ($g_{max}$) less than 200 g, a HIC less than 1000 and a HIC duration ($\Delta t$) greater than 3 ms. The impact area is defined as the area that can be hit by a user after falling through the space in, on or around the playground equipment [23]. The IAS standards were primarily written for testing unitary or rubber-based IAS. Little to no guidance is provided in these standards on the selection of IAS sands. This is fundamentally problematic as an IAS sand can pass the existing $g_{max}$, HIC and $\Delta t$ at the time of installation and weeks later has lost its impact attenuation properties and now represents a serious hazard to children who fall onto it.

The key objectives in this research are:

1. Provide playground designers with confidence that the IAS sands they specify will continue to perform for the intended life of the playground.
2. Provide playground designers with a test method that ensures playground IAS sands are fit-for-purpose.
3. Provide playground designers with guidance on the material properties of playground IAS sands.
4. Provide playground installers with guidance on the selection of IAS sand materials.
5. Provide the suppliers of playground IAS sands with definitive test methods that take account of potential degradation with usage.

## 2. Impact Attenuation Surface Calculations

The head injury criterion (HIC) was first proposed in lieu of the severity index (SI) in the motor vehicle industry as an improved method for measuring the severity of impacts. HIC is a measure of impact severity that considers the duration over which the most critical section of the acceleration pulse persists as well as the peak level of that acceleration. The HIC for each time/acceleration impact is based on the average value of acceleration over the most critical part of the acceleration impulse such that the following function is maximized.

$$\text{HIC} = \left[ \left( \frac{\int_{t_1}^{t_2} a \times dt}{(t_2 - t_1)} \right)^{2.5} \times (t_2 - t_1) \right] \max \tag{1}$$

where:

| | |
|---|---|
| HIC | head injury criterion |
| $t$ | time in milliseconds (ms) |
| $a$ | the acceleration experienced by the headform (missile) and expressed as acceleration due to gravity (g); and |
| $t_1, t_2$ | are two time values which correspond to the start and end of the impact event. |

It can be seen from Equation (1) that the HIC requires the maximization of the mathematical expression involving the time-average acceleration by varying the limits $t_1$ and $t_2$. Note that the acceleration is weighted by the exponent 2.5, and therefore high accelerations for a short time duration will contribute more to the integral than low accelerations for an extended time duration.

The generally accepted maximum HIC value for children within playgrounds is 1000. The loosely correlated value of $g_{max}$ is 200 g. For an ideal IAS, failure would occur when both the HIC and $g_{max}$ simultaneously failed at 1000 and 200, respectively. The rarely occurred as one of these pass/fail criterion always dominates depending on the dynamic properties of the IAS being tested.

It has been suggested by some researchers [24] that the magnitude of the HIC is too high and needs to be lowered to reduce the injury rate and the severity of the injuries, particularly the incidence of long bone injuries. It has been suggested that the impulse force criterion ($I_f$) should be an additional injury prevention criterion for IAS within children's playgrounds [25–27]. The advantage of using $I_f$ as an additional injury criterion is that it considers the bounce momentum that is associated with rubber-based IAS products that have been linked to long bone injuries [28].

## 3. Impact Attenuation Surface Sands

The University of Technology Sydney (UTS) developed a non-laboratory field-based IAS testing device in 1996. In 2000, in collaboration with Green [29], a cableless version of the original device was developed. UTS have used this device, with numerous improvements [30], to perform impact attenuation testing for Australian and New Zealand IAS manufacturers, supervised childcare facilities and local government authorities on a wide variety of IAS. In the period from 1996 to the present, they have performed more than 10,000 impact tests to AS 4422 [6] and the equivalent preceding IAS testing standard [31].

Figure 2 depicts the UTS impact test rig during field measurements on sand within a school playground on the Eyre Peninsula in South Australia.

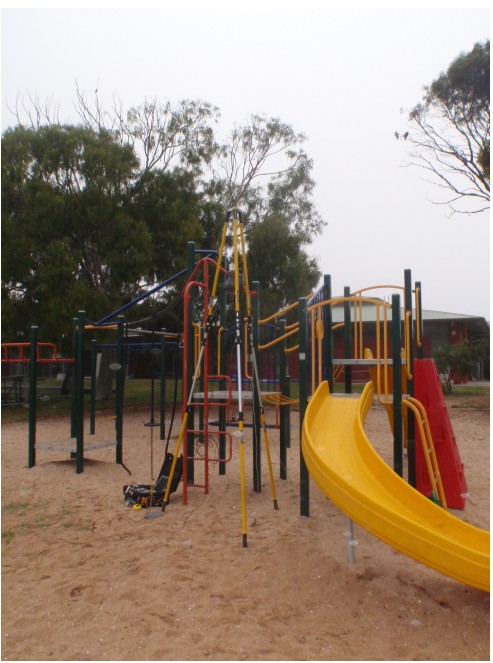

**Figure 2.** UTS IAS testing system. The system has several components including: headform with 3 Endevco piezo-resistive accelerometers mounted at the centroid; a 5 m adjustable tripod; a magnetic release mechanism; and a PC-based data acquisition device sampling at 25,000 Hz.

### 3.1. Force-Displacement

Figure 3 presents force-displacement impact data plots of four difference IAS sands. The force-displacement plot is a useful tool for measuring the likelihood of a severe injury as the envelop of the plot represents the energy absorbed by the sand. The displacement is the penetration of the 4.6 kg instrumented hemispherical headform into the IAS being tested.

It is important that sands which are used in playgrounds can attenuate falls above the maximum free height of fall (equipment height) when they are wet or dry. It is reasonably foreseeable that children will play after rain when the equipment surfaces are still wet and slippery. If the equipment elevated surface is wet the children are more likely to slip on this surface and possibly fall from heights of 3 m onto the sand below.

Figure 3 has six plots of four different IAS sands (two fine and two coarse). The four sands were tested both wet and dry in accordance with AS 4422 [6]. The two coarse sands showed little to no change in their impact attenuation performance and only the results of the dry sand are presented in Figure 3. Whereas, for the fine sands both exhibited a change between the wet and dry tests.

For the wet fine sand data depicted by the brown dotted trace a peak force above 1600 N can be observed indicating poor impact attenuating properties. For the other wet fine sand the peak force was 25% at approximately 400 N. Both these fine sands performed well when were tested in a dry state. The dry test results for these two fine sands are shown by the grey and orange traces where the headform displacement into the sands were 68 mm and 74 mm, respectively. The impact pulse versus time plot for the fine sand that performed well when dry and failed when wet is shown in Figure 4.

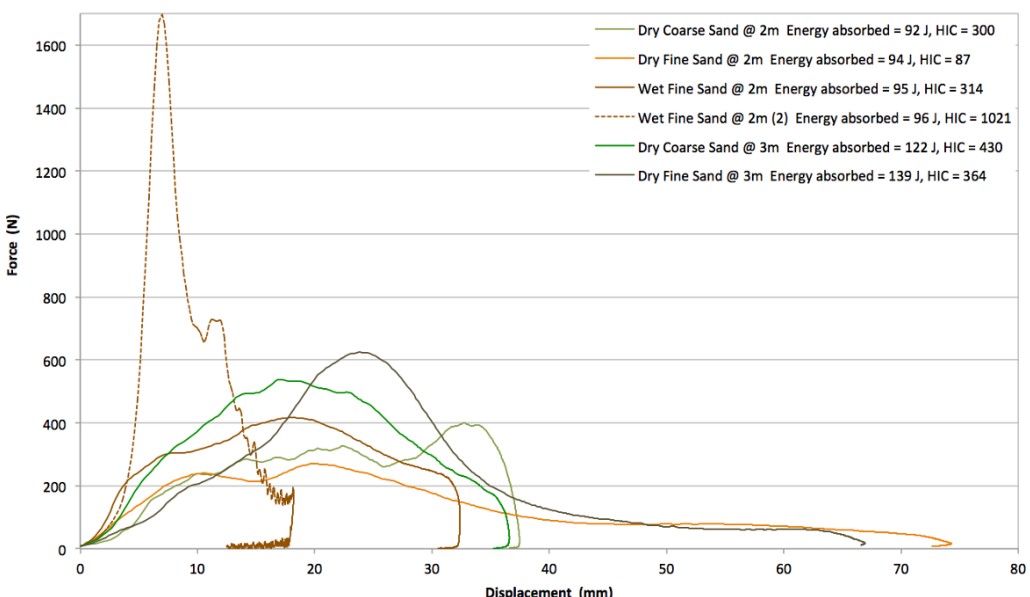

**Figure 3.** The force-displacement curves of six impact attenuation tests conducted using the UTS IAS testing device [32].

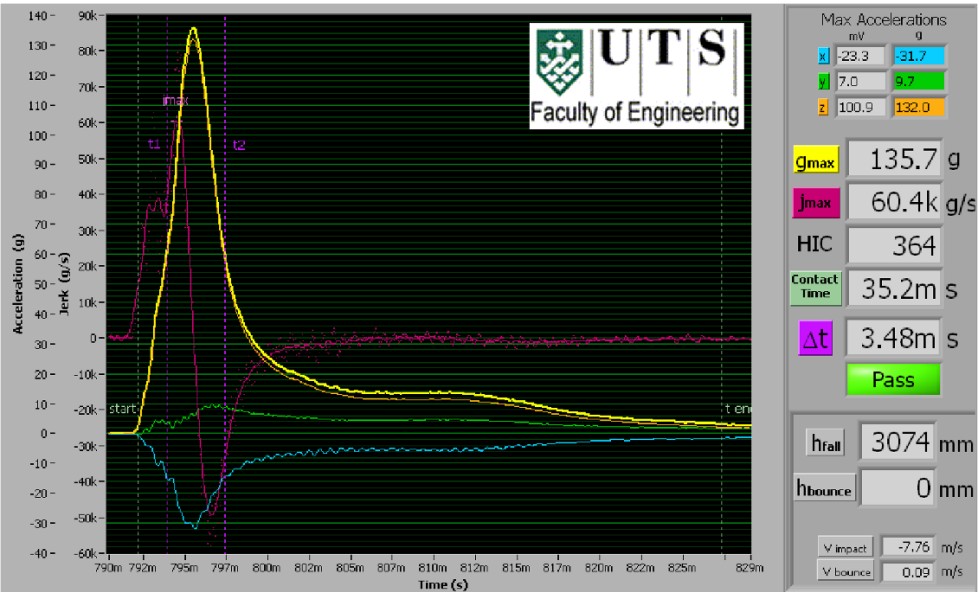

**Figure 4.** IAS sand for a 3 m free height of fall for a fine sand that performed well when dry and failed when wet. The force-displacement curve plot for this sand (Dry Fine Sand @ 3 m Energy absorbed = 139 J, HIC = 364, within Figure 3).

### 3.2. High vs. Poor Performance IAS Sand—Typical Examples

Figures 5 and 6 are provided to show the differences between a high performance IAS and a poor performance IAS sand.

Figure 5 is the impact pulse versus time plot of a typical high performance IAS sand (Sand 04). It shows the asymmetrical $g_{max}$ plot (yellow trace) that is characteristic of sands (rather than the symmetrical plots that are observed with rubber and organic IAS). The plot has a steep leading edge and a more gradual trailing edge. The more gradual trailing edge occurs because the sand particles are flowing away from the impact site and thus allow the impact energy to dissipate. With a $g_{max}$ of 76.4 g (fail = 200 g) and a HIC of 218 (fail = 1000) it greatly exceeds the minimum performance requirements for use in children's playgrounds. This test was also performed 0.5 m above the maximum allowable

playground equipment height (3.0 m). For this sand at 1000 impacts the coefficient of uniformity was less than 2.5, the fineness modulus was 3.16 and percentage fines was 1.0.

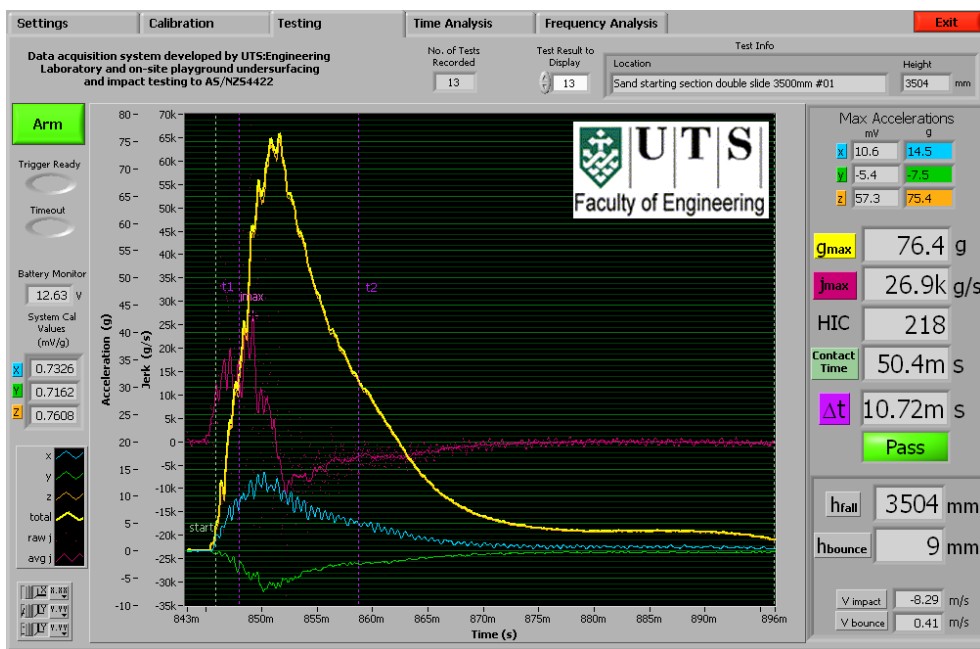

**Figure 5.** High performing IAS sand for a 3.5 m free height of fall. This is the impact pulse versus time plot for impact 1 within Table 1 .

**Table 1.** High performance IAS sand: Four successive impacts in the same location for a 3.5 m free height of fall. This sand passed and provided a critical fall height greater than 3.5 m where the free height of fall (equipment height) was only 2.2 m. Thus, providing the Hobart City Council with a safety margin of more than 1.3 m.

| Impact | $g_{max}$ (g) | $j_{max}$ (g/s) | HIC | $\Delta t$ (ms) | $t_{end} - t_{start}$ (ms) |
|--------|---------------|------------------|-----|------------------|-----------------------------|
| 1 | 76 | 26,900 | 218 | 10.7 | 50.4 |
| 2 | 178 | 164,000 | 569 | 4.4 | 21.2 |
| 3 | 106 | 53,600 | 392 | 9.6 | 23.6 |
| 4 | 173 | 87,000 | 665 | 4.3 | 19.6 |

Figure 6 is the impact pulse versus time plot of a typically poor performance IAS sand. It is the test results of calcareous sand that was being used as an IAS to protect playground equipment with a free height of fall of 3.0 m. It shows a 1469 HIC and a 201$g_{max}$ fail (yellow trace). It also shows both a steep leading edge and an almost vertical trailing edge for the acceleration pulse. This is indicative of little to no sand particle movement and energy dispersion away from the impacted area. This is an undesirable characteristic for children's playgrounds as it means the child is effectively impacting a rigid surface rather than one where the surface is semi-fluid and allows the energy to flow away from the impacted area.

The near vertical trailing edge in Figure 6 is not typical of sand that provides fall attenuation. A typical IAS sand has a steep leading edge with a less steep trailing edge, as depicted in Figure 5 .

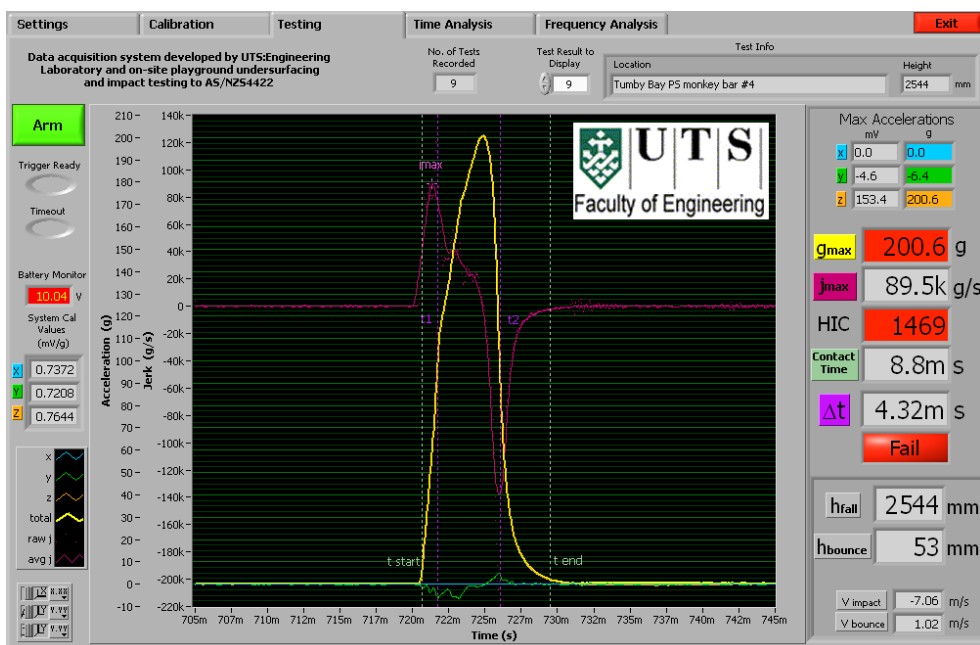

**Figure 6.** Poor performance IAS sand for a 2.5 m free height of fall. This is the impact pulse versus time plot for impact 4 within Table 2.

**Table 2.** Poor performing IAS sand: Four successive impacts in the same location at a 2.5 m free height of fall. The red indicates HIC failures on impacts 2, 3 and 4 together with a $g_{max}$ failure on impact 4. The progressive increase in the $g_{max}$ from impact 1 to 4 shows the material consolidation.

| Impact | $g_{max}$ (g) | $j_{max}$ (g/s) | HIC | $\Delta t$ (ms) | $t_{end} - t_{start}$ (ms) |
|--------|---------------|-----------------|-----|-----------------|----------------------------|
| 1 | 99 | 46,300 | 344 | 5.2 | 29.6 |
| 2 | 192 | 154,400 | 1448 | 4.2 | 8.0 |
| 3 | 187 | 152,000 | 1511 | 4.1 | 8.5 |
| 4 | 201 | 89,500 | 1470 | 4.3 | 8.8 |

## 4. Sand Degradation Analysis

To assess the performance requirements of playground IAS sand several additional performance requirements beyond those required by AS 4422 [6] and its equivalent standards were considered, namely: sand particle distribution, grading curves, sand particle shape, coefficient of uniformity, fineness modulus, percentage fines, soluble and semi-soluble salts, and sand particle degradation. These parameters will now be explained in more detail.

### 4.1. Sand Particle Distribution

A representative portion of sand (subject to zero degradation) should be subject to mechanical sieve analysis using the wet sieve process described within the Australian Standard AS 1141.11.1 [33] or its equivalent. It has been observed that to be an appropriate sand for impact attenuation in children's playgrounds, the coefficient of uniformity must not exceed 2.75. This is intended to serve as a maximum acceptable criterion. It is recommended that the desired value should be less than 2.00. It is worth noting that playground sand can be contaminated by the mixing of two tested certified products that have different particle size distributions. For example, if fine sand is mixed with coarse sand while topping up the level during routine maintenance the impact attenuation properties will be significantly changed by filling the inter-particle spaces and thus lower or prevent the particle flow when the child falls from the playground equipment.

### 4.2. Grading Curves

To understand the particle size distributions of different sands considered within this investigation, mechanical sieve analysis processes were used to divide a sample portion of material into like particle sizes. An appropriate way of presenting the results of such an exercise is on a particle size distribution grading curve—plotting the cumulative percentage of material passing as against a given sieve size. A well graded sand has a reasonable proportion of the particles distributed over a range of sizes while a poorly graded sand has the majority of particles within a narrow size range (see Figure 7).

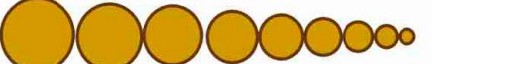 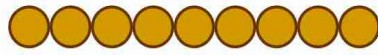

**Figure 7.** Sand grading: (**Left**) Well graded. (**Right**) Poorly graded.

The objective of an IAS sand is to cushion the fall. It was suggested that to minimise the impact forces and the energy transformation associated with the fall, this needs to occur slowly and progressively, over the longest possible time duration and having the greatest penetration into the IAS. One may therefore conclude that it is desirable for sand to be in a loosely packed state such that particles can move to different positions upon impact [34].

Accordingly, it is expected that the particle size distribution of sand will have direct and measurable effects on its impact attenuating performance. Figure 8 shows for well graded materials the small particles typically fit into the voids while for poorly graded materials there are no particle sizes able to fill the voids. Impact testing confirmed that well graded products did not perform well when tested to AS 4422 while the poorly graded sands provided much lower HIC and $g_{max}$ results thus making them desirable for use as IAS sands within playgrounds.

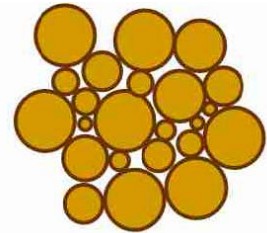 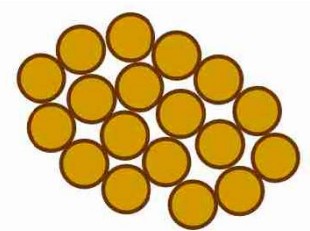

**Figure 8.** Sand grading: (**Left**) Well graded tightly packed. (**Right**) Poorly graded loosely packed.

### 4.3. Sand Particle Shape

Particle shape is important for impact attenuating sand. The energy of the falling child's impact needs to be dissipated by the sand particles moving away from the impact site. The shape of the sand particle that allows this movement is a 'well rounded' and 'high sphericity' shaped particle as defined in Figure 9. Moreover, this type of sand will prevent particles interlocking hence forming a desirable loose IAS condition [35]. A representative portion of sand (subject to zero degradation) should be analysed through microscopic investigation—at approximately ten times magnification. The particle shape should then be assessed considering the sphericity and roundness using Figure 9.

| Roundness classes | Very Angular | Angular | Sub-angular | Sub-rounded | Rounded | Well Rounded |
|---|---|---|---|---|---|---|
| High Sphericity | | | | | | |
| Low Sphericity | | | | | | |
| Roundness indices | 0.12 to 0.17 | 0.17 to 0.25 | 0.25 to 0.35 | 0.35 to 0.49 | 0.49 to 0.70 | 0.70 to 1.00 |

**Figure 9.** Chart for estimating the roundness and sphericity of sand particles [36].

### 4.4. Coefficient of Uniformity

The coefficient of uniformity is applied within engineering and is calculated as the particle size where 60% of the material is passing, divided by the particle size where 10% of the material is passing—interpolated from the grading curve. For poorly graded sand the coefficient of uniformity should not be greater than 6.00 [37], while for an ideal IAS sand, a preferable range of the coefficient of uniformity is less than 2.00 as mentioned earlier in Section 4.1. Figure 10 depicts three sand types, each with a different particle size but poorly graded.

This study found that size did not matter provided the particles within a particular sand were all the same size relative to each other. A corollary to this finding was that a high performing IAS sand can be contaminated by topping up with an equally high performing IAS with a different particle size. This is because the smaller sand will occupy the voids between the larger sand and reduce the inter-particle movement.

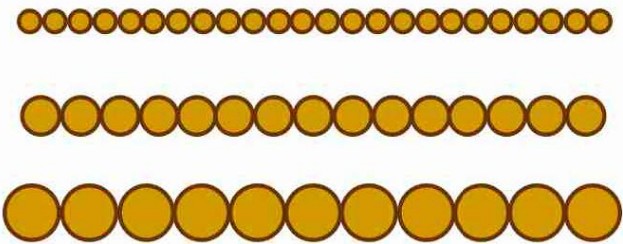

**Figure 10.** Like size particles equates to a low coefficient of uniformity and better IAS characteristics.

Figure 11 is a *manufactured* IAS sand. This sand was manufactured by passing a 'high sphericity' and 'well rounded' granite river sand through a pair of under/over sieves and collecting the sand that did not pass the 2.75 mm sieve, but did pass the 4.00 mm sieve. When installed the manufactured sand was for all intents and purposes *self-levelling*. When you walked across the installed sand your foot sank below the ankle and your shoe filled with sand. When you lifted your foot the indentation left by your shoe was filled up as the sand particles rolled into the indentation. The other notable property was that you could conduct 100 or more consecutive impact tests and there was no measurable increase in either the $g_{max}$ or HIC values. There was no compaction, quite a remarkable sand.

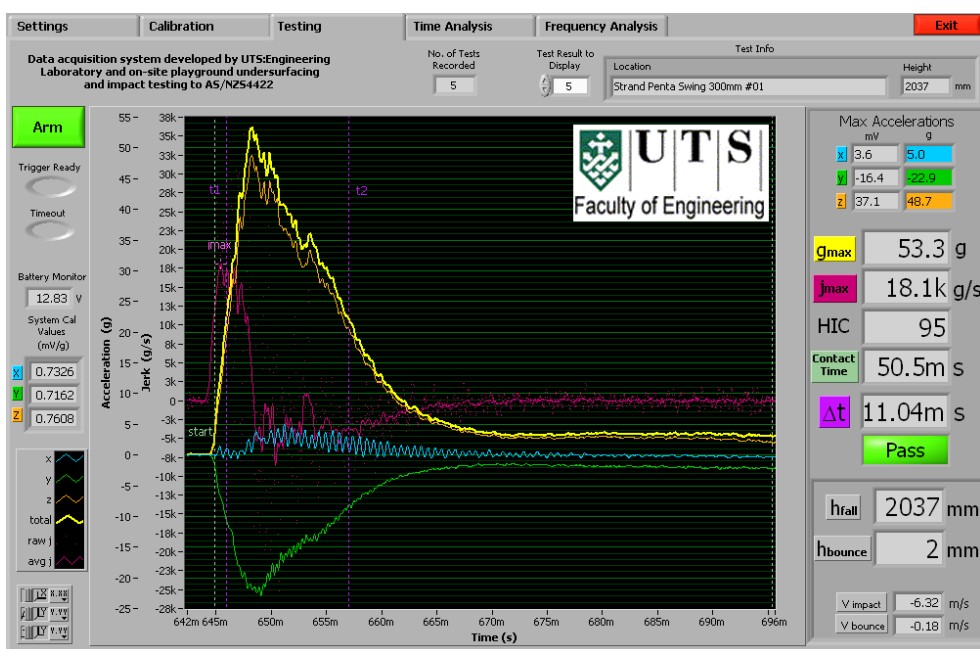

**Figure 11.** High performing *manufactured* IAS sand for a 2.0 m free height of fall.

### 4.5. Fineness Modulus

A key limitation of the coefficient of uniformity is that it fails to provide any indication of a sand's approximate particle size—it only indicates whether particles are of similar or different sizes. As an example, the three gradings indicated in Figure 10 would yield the same coefficient of uniformity—but they are notably different in terms of particle size.

Therefore, a second indicator has been employed to mathematically indicate the fineness or coarseness of particle size—the fineness modulus.

Neville and Brooks [38] describe this as the sum of the cumulative percentages of material retained on the sieves of the standard series, divided by 100. It must be appreciated that the standard series consists of sieves with aperture size twice that of the preceding sieve—for this project 0.075, 0.15, 0.3, 0.6, 1.18 and 2.36 mm. This modulus is used to indicate the approximate particle size of a sand sample and thereby addresses the failure of the coefficient of uniformity to do so.

### 4.6. Percentage Fines

Measurements have shown that to be appropriate sand for impact attenuation in children's playgrounds the percentage of material passing a 0.075 mm sieve aperture must not exceed 1.4%. This value is intended to serve as an absolute maximum acceptable criterion. It is recommended that a value as close to 0.0% is desirable. It is worth noting that sand can be contaminated after installation beneath the playground equipment. The two major causes of fines contamination are runoff and wind-blown dust particles. Thus, it is important when planning to install a playground that the impact attenuating material and the site are appropriately matched. For example, it would not be appropriate to install impact attenuating sand in a region subject to frequent dust storms; or at the base of a clay embankment subject to runoff.

### 4.7. Soluble and Semi-Soluble Salts

It has been observed that soluble and semi-soluble salts can act as binding agents in playground IAS sands. Soluble salts can be introduced to playgrounds by wind carrying sea spray, using bore water, or run-off contaminants. Soluble or semi-soluble salts were not evaluated during degradation testing in the current study. This will be considered in a future study.

*4.8. Sand Particle Degradation*

During the experimental phase of this project, it was noted that sands that previously had exhibited excellent impact attenuation properties deteriorated with time. On closer inspection, it was established that these sands were degrading with usage. The sand particles were grinding against one another creating fines. The following test was developed as a simple and effective method to measure the observed degradation with usage. It is effectively an accelerated ageing test that employs a commonly used civil engineering compaction test.

The test method is defined within the Australian Standard AS 1289.5.1.1 Methods of testing soils for engineering purposes [39]. Simply, the sand sample is positioned within a cylindrical steel container. It is then repeatedly compacted by the impacts from a ram that is allowed to fall freely through a specified drop height. AS 1289.5.1.1 is intended primarily to test material compaction. However, Carter and Bentley [20] identify a key failing of the intended test when applied to sands and gravels. They noted sand and gravel tended to be displaced by the impact of the ram, as opposed to being compacted. The net effect was that degradation/deterioration occurred. They noted that the extent of this degradation could be quantified through mechanical sieve analysis. This observed behaviour made the AS 1289.5.1.1 test method ideal for testing the degradation of playground impact attenuating sands. Three equal and representative portions of sand should be subject to the AS 1289.5.1.1 Compaction effort methodology [39]. It is recommended that 1 kg samples should be subjected to 1500 impacts from the 2.7 kg ram falling through a free height of fall equal to 300 mm. Thereafter, material portions should be mechanically sieved using the wet sieve process described by AS 1141.11.1 [33].

## 5. Assessment of 15 Different Sands

Fifteen different sands were evaluated in this study. The samples have been de-identified so as to anonymise the names, locations and contact details of the various playgrounds and sand suppliers. Sands 01 to 06 and 12 were all assessed as passing AS 4422 [6] with a 2.5 m or greater critical fall height when installed at a depth of 300 mm or greater. They were all marketed and sold as IAS sands. The remaining five sands were not marketed or sold as IAS sands. Sands 07 and 08 were beach sands while Sands 09 to 11 were engineering sands (concreting, bricklaying and paving). Each sand sample was subjected to accelerated degradation using the Los Angeles Abrasion Test. The test is defined under the Australian Standard AS 1141.25.3—2003 Methods for sampling and testing aggregates: Degradation factor—Fine aggregate [40]. The detailed specifications of these sands regarding the coefficient of uniformity, fineness modulus, % fines, and degradation after 0 and 1500 impacts are listed in Table 3. Note that most of the sands can be classified as poorly graded sand (coefficient of uniformity >5) except Sands 11 and 12A. Among those poorly graded sands, Sands 06 to 08 are preferable with a coefficient of uniformity close to 2. The results of Sands 06 and 07 stand out as their % fines value is close to zero.

Figures 12–19 show the grading curves and the $10\times$ magnification of the sand before impact loading from selected sands. For each sand, four grading curves after 0, 500, 1000, and 1500 impacts are given. Grading curve shifting left after impact loading indicates particle degradation occurs during impact [41]. It is worth noting that Sand-06 (high sphericity and rounded) shows negligible degradation after 1500 impacts, Sand-10 and Sand-11 present significant degradation (>7) after 1500 impacts, others show relatively medium degradation (2.3 to 3.6). Therefore, combining all the above information, Sand 06 is the most ideal IAS among all the sand samples reviewed.

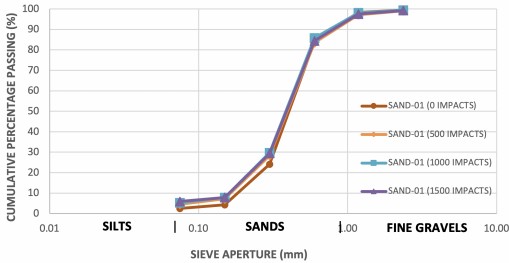
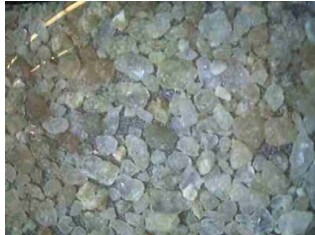

**Figure 12.** Sand 01: (**Left**) Similar grading curves for 500, 1000 and 1500 impacts, indicating little degradation. (**Right**) 10× magnification at 0 impacts depicting medium sphericity and rounded.

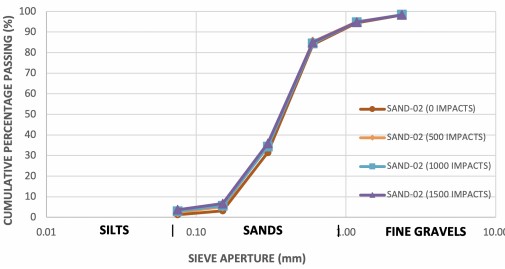
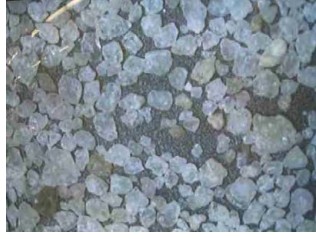

**Figure 13.** Sand 02: (**Left**) Similar grading curves for 500, 1000 and 1500 impacts, indicating very little degradation. (**Right**) 10× magnification at 0 impacts depicting high sphericity and rounded.

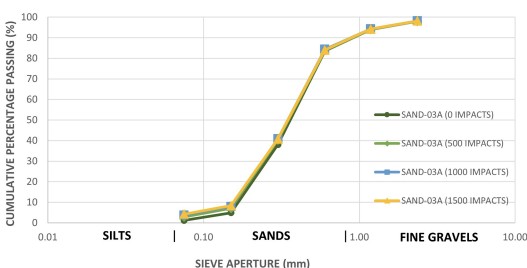
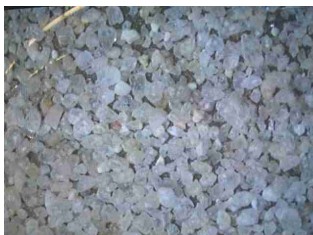

**Figure 14.** Sand 03A: (**Left**) Similar grading curves for 500, 1000 and 1500 impacts, indicating very little degradation. (**Right**) 10× magnification at 0 impacts depicting high sphericity and rounded.

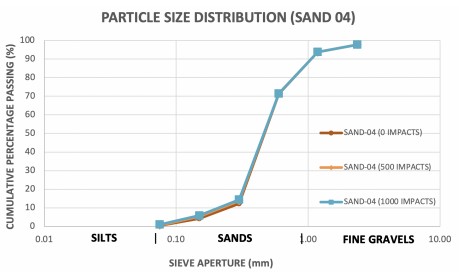
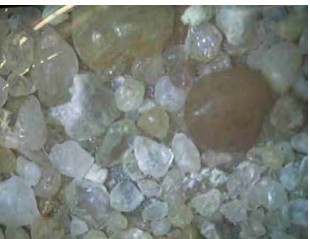

**Figure 15.** Sand 04: (**Left**) Similar grading curves for 500 and 1000 impacts, indicating very little degradation. (**Right**) 10× magnification at 0 impacts depicting high sphericity and well rounded. Critical fall height is greater than 3.5 m when tested in accordance with AS 4422:2016 (see Figure 5).

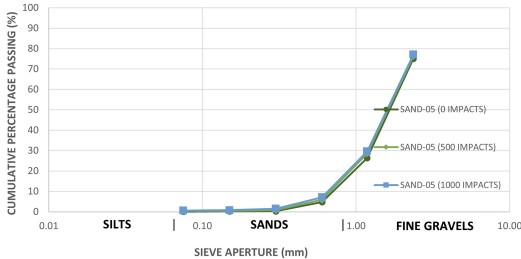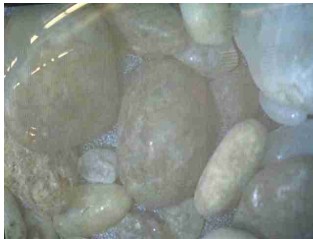

**Figure 16.** Sand 05: (**Left**) An example of a sand where the sand particles were larger than sands traditionally used within playgrounds. This type of sand is more commonly used for potting as it provides excellent drainage. Similar grading curves for 500 and 1000 impacts, indicating very little degradation. (**Right**) 10× magnification at 0 impacts depicting high sphericity and well rounded.

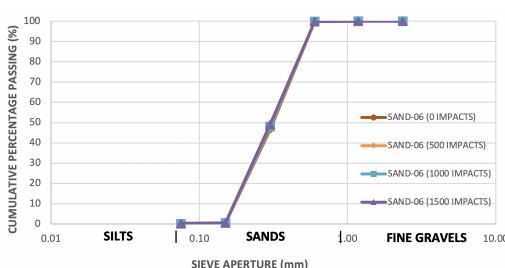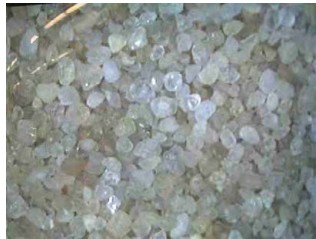

**Figure 17.** Sand 06: (**Left**) Similar grading curves for 500, 1000 and 1500 impacts, indicating very little degradation. (**Right**) 10× magnification at 0 impacts depicting high sphericity and well rounded.

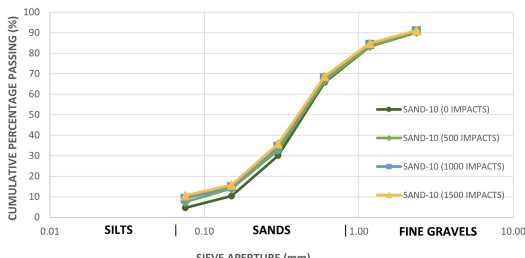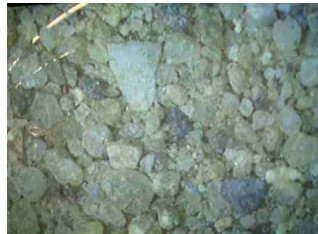

**Figure 18.** Sand 10: A typical bricklayer's sand not suitable as an IAS for children's playgrounds. (**Left**) Slight separation in the grading curves for 500, 1000 and 1500 impacts, indicating little degradation. (**Right**) 10× magnification at 0 impacts depicting medium sphericity angular.

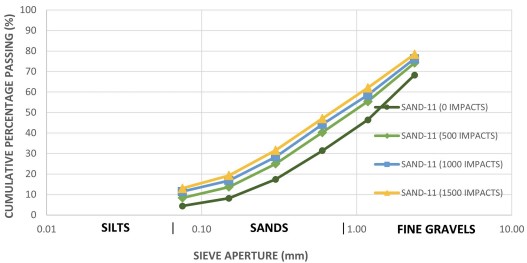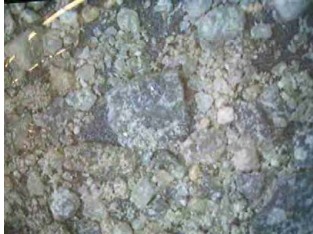

**Figure 19.** Sand 11: A typical paving sand not suitable as an IAS for children's playgrounds. (**Left**) Separation in the grading curves 500 and above impacts, indicating little degradation. (**Right**) 10× magnification at 0 impacts depicting low sphericity sub-rounded.

**Table 3.** Summary table for all 15 sands analysed (* 1000 impacts).

| Sand | Coefficient of Uniformity | Fineness Modulus | % Fines | Degradation 0 Impacts | Degradation 1500 Impacts |
|------|---------------------------|------------------|---------|-----------------------|--------------------------|
| 01 | 2.45 | 2.87 | 2.5 | 2.45 | 2.60 |
| 02 | 2.40 | 2.87 | 1.3 | 2.40 | 2.51 |
| 03A | 2.36 | 2.80 | 1.2 | 2.36 | 2.50 |
| 03B | 2.33 | 2.71 | 1.6 | 2.33 | 2.38 |
| 03C | 2.31 | 2.72 | 1.3 | 2.31 | 2.34 |
| 04 | 2.10 | 2.20 | 0.4 | 2.10 | 2.47 * |
| 05 | 2.75 | 2.93 | 0.1 | 2.37 | 2.80 * |
| 06 | 2.03 | 2.53 | 0.2 | 2.03 | 1.98 |
| 07 | 2.09 | 2.70 | 0.4 | 2.09 | 2.19 |
| 08 | 1.97 | 1.91 | 1.2 | 1.97 | 3.57 |
| 09 | 3.47 | 3.39 | 0.8 | 3.47 | 2.48 |
| 10 | 3.55 | 3.16 | 4.6 | 3.55 | 7.04 |
| 11 | 10.39 | 4.24 | 4.4 | 10.39 | 20.00 |
| 12A | 8.22 | 2.66 | 14.2 | - | - |
| 12B | 5.68 | 2.69 | 8.8 | - | - |

## 6. Conclusions

This paper proposed an additional IAS test to eliminate sands that degraded above an established threshold rate after installation due to normal usage. IAS degradation properties of fifteen sands were tested including sand particle shape, sand particle distribution, percentage fines and sand particle degradation. This accelerated ageing test method would apply only to sands and not rubber and wood fibre IAS products.

It is recommended that for sand to be used for impact attenuation within children's playgrounds, the following criteria must be satisfied:

1. The coefficient of uniformity must not exceed 2.70.
2. The percentage of material passing a 0.075 mm sieve aperture must not exceed 3.1% when compared to the value ascertained for percentage fines.
3. The percentage fine material passing a 0.075 mm sieve is less than 4.5%.

Nevertheless, a coefficient of uniformity less than 2.00 together with percentage fines approaching 0.00 is preferred.

The following conclusions are drawn from this study:

- Selected sand can possess excellent long-term impact attenuation properties that make it ideal for usage within children's playgrounds.
- Sand with a particle shape that is rounded to well-rounded has excellent impact attenuation properties which makes it ideal for usage in children's playgrounds where it enables the energy to flow and disperse away from the point of impact.
- High levels of fines and clay within playground sand are correlated with poor impact attenuation as these tend to bind the sand particles and reduce the flow and dispersion of energy away from the point of impact.
- IAS sand used within children's playgrounds can and does degrade with usage and this can be dramatically reduced by better material selection.
- The playground sand particle size is not particularly important.
- The grading coefficient of playground sand is important and the coefficient of uniformity should be kept below 2.75.
- The early identification of playground sand types that exhibit robust resistance to degradation during usage should be encouraged.
- The installation of playground sand types that exhibit non-robust stability to degradation (generation of clay and fines) during usage should be discouraged and/or eliminated.

In addition to these conclusions, it was noted that the best IAS sands were sourced from quarries located on rivers that had eroded volcanic outcrops. These sands were shown

to degrade the least and their particle shape was rounded to well-rounded. The most reliable location for good quality IAS sands on these rivers was on specific bends. The sand mined at these locations consistently had a tight particle size distribution.

Sand classified as unsuitable for use as IAS sand for children's playgrounds was sourced from calcareous rock, coral and seashell middens.

**Author Contributions:** All authors contributed equally to this publication. All authors have read and agreed to the published version of the manuscript.

**Funding:** This research was internally funded by the University of Technology Sydney.

**Institutional Review Board Statement:** Not applicable.

**Informed Consent Statement:** Not applicable.

**Data Availability Statement:** The data presented in this study are available on request from the corresponding author.

**Acknowledgments:** The authors acknowledge support and data from Daniel Weaver, Warwick Howse and various Australian local government authorities.

**Conflicts of Interest:** The authors declare that there is no conflict of interest.

## Abbreviations

The following abbreviations are used in this manuscript:

| | |
|---|---|
| $g_{max}$ | maximum acceleration |
| HIC | head injury criterion |
| IAS | impact attenuating surface |
| $I_f$ | impulse force criterion |
| $j_{max}$ | maximum jerk |
| $t_{end} - t_{start}$ | Duration of the impact contact time |
| $\Delta t$ | Duration of the HIC |

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
