# Peer review of "Additional Criteria for Playground Impact Attenuating Sand"

_applsci, doi:10.3390/app11198805_

Round 1
Reviewer 1 Report
The paper does not present data, it appears to present option and future tests that needs to be presented. Only data presented are grading curves for arbitrarily named sands . It does not even show how each sand is different from the others based on either particle shape or density or composition. A lot of information is missing and needs to be improved heavily. For the only graphs displayed the term " % of passing is not defined"
Author Response
Thank you for your comments.
The paper has been rewritten and content rearranged into a more logical order.
The Introduction has been expanded and a literature review added. The total number of citations has been increased from 18 to 41. The introduction now includes more background on the research problem that the paper aims to address.
Background to the HIC has been added and the If removed.
The discussion on additional performance variables was revised.
The grading curves, together with 10x magnification photographs of each sand are now included in the manuscript.
The flowchart has been removed.
The conclusions have been rewritten to reflect the rewritten paper.
A copy of the revised paper is attached.

Reviewer 2 Report
Dear Authors,
the paper does not possess the minimum acceptable quality level for publication, since it does not present sufficient experimental data quality and appropriate discussion.
Below are reported the critical points that must be clarified to raise the minimum quality level requested for considering publication:
1) General paper organization - The work is proposed by the Authors as a research article; however, it appears to be a “mix” between a research article (minor part) and a review article (with a too low number of citations). While the Introduction is extremely short, as well as the Experimental results, Sections 4 and 5 are well developed, but these parts present almost general considerations deriving from literature data.
2) Regarding the Introduction (Section 1), it needs to be expanded to define the problematic better, also citing additional works besides standards and self-citations.
3) Section 2 presents how the HIC is calculated, but the origin of the proposed equation is not declared. Please do it. Then, the Authors also present the equation for calculating the impulse force criterion (If). However, since HIC and If are not presented in the Experimental results, the reader does not understand why such equations are provided.
4) The Experimental results section (Section 3) provides the so-called grading curves for sands. As stated in the abstract and at the beginning of the present section, fifteen type of sands were analysed, but only five ones are discussed. All the tested sands need to be discussed, while the current discussion appears to be too short. Moreover, the Authors do not present pieces of evidence about the morphology of the sand before and after the test (e.g. optical micrographs) for evaluating the critical parameters listed in Section 4: particle shape, particle distribution and percentage fines. For these reasons, the experimental part does not have an acceptable quality level.
5) Section 4 is almost a state-of-the-art part, with some little and unclear reference to the results obtained in the present work. However, no proofs are provided for such affirmations, like the following “No soluble or semi-soluble salts are present or produced during the degradation testing.” (lines 128-129, page 6).
6) Section 5 aims at providing a flow chart for selecting sand. Again, this chart is literature data only.
7) The Conclusions (Section 6) resume some of the most important characteristics that sand should possess to have a good impact attenuation capability, but the Authors do not discuss the results of the experimental work appropriately (or without supporting experiments).
Author Response
Response to Reviewer 2
The paper does not possess the minimum acceptable quality level for publication, since it does not present sufficient experimental data quality and appropriate discussion.
General paper organization - The work is proposed by the Authors as a research article; however, it appears to be a “mix” between a research article (minor part) and a review article (with a too low number of citations). While the Introduction is extremely short, as well as the Experimental results, Sections 4 and 5 are well developed, but these parts present almost general considerations deriving from literature data.
Thank you for your comments.
The paper has been rewritten as a research article and now addresses your comments.
The number of citations has been increased from 18 to 41.
The introduction has been expanded from approximately 1 page to approximately 2.5 pages.
A copy of the revised paper is attached.
Your comments are addressed in detail below.
Comment 1
Regarding the Introduction (Section 1), it needs to be expanded to define the problematic better, also citing additional works besides standards and self-citations.
Response 1
Section 1 has been expanded and now includes a more detailed background to the research problem that this paper aims to address. The number of citations has been increased from 18 to 41 as stated above.
Comment 2
Section 2 presents how the HIC is calculated, but the origin of the proposed equation is not declared. Please do it. Then, the Authors also present the equation for calculating the impulse force criterion (If). However, since HIC and If are not presented in the Experimental results, the reader does not understand why such equations are provided.
Response 2
The origin of the HIC is now declared. The source of the equation is also referenced. Data for the gmax, jmax, HIC, delta-t and tend - tstart are presented in Figures 1, 2 and 7, they are also presented in Tables 1 and 2.
Additional discussion is also provided.
The If equation has been deleted as it is not directly relevant to the discussion herein.
Comment 3
The Experimental results section (Section 3) provides the so-called grading curves for sands. As stated in the abstract and at the beginning of the present section, fifteen type of sands were analysed, but only five ones are discussed. All the tested sands need to be discussed, while the current discussion appears to be too short. Moreover, the Authors do not present pieces of evidence about the morphology of the sand before and after the test (e.g. optical micrographs) for evaluating the critical parameters listed in Section 4: particle shape, particle distribution and percentage fines. For these reasons, the experimental part does not have an acceptable quality level.
Response 3
The experimental results have been moved to Section 5. The results for all 15 sands are now presented together with 10x magnification photographs of each sand.
Comment 4
Section 4 is almost a state-of-the-art part, with some little and unclear reference to the results obtained in the present work. However, no proofs are provided for such affirmations, like the following “No soluble or semi-soluble salts are present or produced during the degradation testing.” (lines 128-129, page 6).
Response 4
This statement and reference to ‘no soluble or semi-soluble salts’ has been removed from the manuscript because this parameter was not tested.
Comment 5
Section 5 aims at providing a flow chart for selecting sand. Again, this chart is literature data only.
Response 5
The flow chart has been deleted to avoid confusion.
Comment 6
The Conclusions (Section 6) resume some of the most important characteristics that sand should possess to have a good impact attenuation capability, but the Authors do not discuss the results of the experimental work appropriately (or without supporting experiments).
Response 6
The conclusions have been rewritten to reflect the rewritten paper.

Round 2
Reviewer 2 Report
Dear Authors,
the revised paper has been significantly improved, and now possesses an acceptable quality level. For this reason, I advise the publication after the correction of the optical images in Figures 18 and 19, which are identical in the current draft.
Author Response
Thank you for finding this coding error within the LaTex file.
I have corrected the attached file.
